# ESCAViT: Symmetry-Aware EEG Classification

Se Hwan Lim[1,2][0009−0006−5943−3902] and Hyun Gyu Lee[3,4,*]

[1]Department of Electronic Engineering, Inha University, Republic of Korea
[2]Department of Artificial Intelligence Semiconductor Engineering, Inha University, Republic of Korea
[3]Department of Electrical and Computer Engineering, Inha University, Republic of Korea
[4]College of Medicine, Inha University, Republic of Korea
`tpghks726@inha.edu, hglee@inha.ac.kr`

**Abstract.** Accurate classification of Ictal-Interictal-Injury Continuum (IIIC) patterns is essential for neurological assessment in intensive care units, yet remains challenging due to limitations in capturing inter-lead correlations and addressing class imbalance. To tackle this, we propose ESCAViT, a multi-stream Transformer-based EEG classification framework. ESCAViT leverages the Video Vision Transformer with specialized feature extraction mechanisms to model spatiotemporal EEG patterns, while applying domain-adaptive learning to enhance data diversity and mitigate heterogeneous Other class(HOC) effects. Experimental results on the IIIC dataset show that ESCAViT outperforms state-of-the-art models, achieving **21.9%** improvement in mean accuracy per class (mACC) and **22.6%** in F1-score. Our method significantly enhances LRDA classification by over **over 20%**, thereby addressing classification bias. ESCAViT demonstrates consistent performance across different IIIC patterns and imbalanced distributions, confirming its effectiveness in EEG classification. The code is available at https://github.com/limshmai/ESCAViT.git

**Keywords:** IIIC Pattern Classification · EEG Transformer · Inter-Lead Contrastive Learning · Class Imbalance Mitigation.

## 1 Introduction

Electroencephalogram (EEG) monitoring plays a vital role in detecting and managing neurological injuries in intensive care units (ICUs) [1]. Among various EEG patterns, IIIC patterns are frequently observed in critically ill patients. These patterns, which include Seizure, Lateralized Periodic Discharges (LPD), Generalized Periodic Discharges (GPD), Lateralized Rhythmic Delta Activity (LRDA), and Generalized Rhythmic Delta Activity (GRDA), provide crucial diagnostic insights into subclinical seizures and seizure-like electrical events, aiding early neurological injury detection [2].

However, IIIC pattern classification remains challenging due to two key factors:

1. Inter-lead relationships & spatial dependencies—In IIIC classification, Lateralized patterns are confined to one hemisphere, while Generalized patterns manifest symmetrically across both hemispheres. This hemispheric symmetry is crucial for distinguishing seizures from non-ictal activity, yet existing methods fail to effectively model these inter-lead correlations [3, 4].
2. Data ambiguity & class imbalance—Expert disagreement and severe class imbalance (Other: 7,205 vs. LRDA: 936 ) introduce significant classification bias [5, 6].

To address these challenges, we propose ESCAViT, a multi-stream Transformer-based EEG classification framework that explicitly models EEG lead symmetry and enhances feature robustness through domain-adaptive learning. Unlike conventional Video Vision Transformer (ViViT) architecture [7], which struggles with local representation, ESCAViT introduces the following key contributions:

1. **Lead-Aware Feature Extraction:** Pairwise Attention and Lead Attention are incorporated to explicitly capture inter-lead dependencies, improving spatial-temporal EEG representation.
2. **Domain-Specific Learning Strategies:** To mitigate class imbalance and HOC issues and enhance feature generalization, we design a unified framework that integrates Adaptive EEG Spectrogram Mixup (AES-Mix) and Lead Interrelation-Guided Contrastive Learning (LIGCL).
3. **Multi-Pathway Feature Integration:** Overlapping Convolutional Projection and Multi-Stream Architecture enable fine-grained seizure pattern detection while preserving global EEG structure.

Through these innovations, ESCAViT significantly enhances IIIC pattern classification, particularly in capturing symmetrical relationships between left and right hemisphere leads, outperforming existing methods.

## 2    Related Work

EEG classification presents unique challenges due to its high-dimensional, noisy, and ambiguous nature, making traditional spectrum analysis and wavelet-based methods suboptimal [8]. Recent deep learning approaches have demonstrated superior performance across various EEG-related tasks, including seizure detection and neurodegenerative disease diagnosis [9].

**Hybrid GNN-CNN Models.**. Hybrid GNN-CNN models [3] and spatial multi-scale attention mechanisms [10] have been introduced to address these limitations. However, these methods still face challenges in modeling dynamic feature interactions due to the structural rigidity of GNN-based graph representations.

**Transformer-Based EEG Analysis.** Transformers leverage self-attention mechanisms for long-range feature extraction [4], yet lack explicit spatial priors, limiting their ability to model local EEG variations and inter-lead correlations. ViT-based architectures [11] show promise in EEG classification but struggle to capture fine-grained seizure morphology changes.

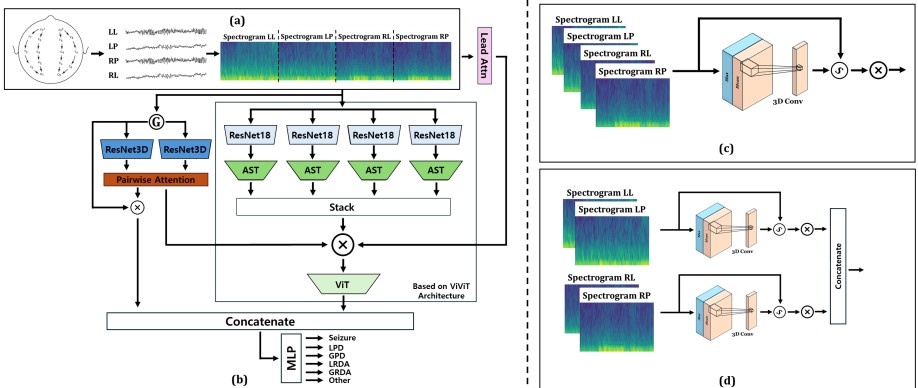

**Fig. 1.** Architecture of ESCAViT. (a) EEG preprocessing: 20 raw leads compressed into four key leads (LL, LP, RL, RP) and converted to Mel-Spectrograms. (b) ESCAViT structure: integration of ViViT-based inter-channel modeling and hemisphere-specific 3D ResNet pathways. (c) Lead-Attention and (d) Pairwise Attention mechanisms for modeling inter-lead relationships and enhancing spatial-spectral feature extraction.

**Motivation for ESCAViT.** Existing EEG models fail to comprehensively address lead symmetry, class imbalance, and data ambiguity. To resolve these issues, we propose ESCAViT, which integrates ViViT-based multi-stream feature learning with domain-specific techniques for effective IIIC pattern classification.

## 3  Methodology

### 3.1  Overview

The preprocessing pipeline of ESCAViT comprises two stages, as illustrated in Fig. 1(a). Initially, the Banana Montage technique [12] reduces the original 20 EEG leads to four key leads (LL, LP, RL, RP), thereby reducing computational complexity while preserving spatial relationships. Subsequently, each lead is transformed into a Mel-Spectrogram [13] with a temporal axis of 256 seconds to facilitate time-frequency analysis. In the lead notation, the first letter denotes the Left/Right hemisphere, while the second letter indicates the Lateral/Parasagittal position.

As shown in Fig. 1(b) and (c), (d), ESCAViT integrates ViViT-based inter-channel modeling with hemisphere-specific spatiotemporal feature extraction to overcome the limitations of conventional ViViT models. The architecture consists of a ViViT-based pathway for global feature extraction and two 3D ResNet pathways that independently learn spatiotemporal EEG representations from each cerebral hemisphere. Convolutional Projection and Lead Attention mechanisms are incorporated to explicitly capture inter-lead dependencies. Additionally, ESCAViT applies a unified framework that integrates AES-Mix for data

augmentation and LIGCL for contrastive learning to improve class separability and robustness against data ambiguity.

### 3.2   Feature Extraction with Lead Attention

Seizure EEG signals are characterized by the sudden appearance of distinct spectral patterns at specific time points. To effectively capture these temporal and spectral fluctuations, we propose a Lead Attention mechanism based on spatial attention [14]. Unlike CBAM, which employs 2D spatial pooling, Lead Attention explicitly models inter-lead dependencies and EEG-specific time-frequency variations while preserving temporal information through 3D convolutions.

Lead Attention dynamically learns the importance of four leads at each time frame. By extracting mean and maximum values from the time-frequency representations of each lead and generating attention weights through 3D convolutions, the mechanism can selectively focus on specific leads exhibiting seizure activity. Pairwise Attention groups left hemisphere leads (LL, LP) and right hemisphere leads (RL, RP) to explicitly model inter-hemispheric symmetry.

For lead-wise feature extraction, the AST architecture employs DeiT-Base (12 layers, 768 dimensions, 12 attention heads) applied to each lead, dividing mel-spectrograms into $16 \times 16$ patches. Unlike standard AST models, ESCAViT incorporates Overlapping Convolutional Projection to overcome the limitations of ViT-based models in capturing fine-grained seizure morphology [15, 16]. As illustrated in Fig. 1(d), Lead Attention extracts mean and maximum values along the frequency and time axes and generates attention weights through a 3D convolutional network. These weights refine the AST-based representations to enhance localized seizure pattern detection.

### 3.3   Feature Integration

Extracted features from each lead are integrated using a Global Feature Transformer (Fig. 1(b)), which is a ViT-Base model pretrained on ImageNet. The Global Feature Transformer integrates four leads as $2 \times 1$ patches and employs learnable absolute position embeddings at all stages. This integration leverages Convolutional Projection [16] for enhanced local feature encoding and overlapping patch embeddings to maintain critical long-range dependencies.

Pairwise Attention (Fig. 1(c)) models hemispheric relationships by distinguishing left-right asymmetries, thereby improving seizure pattern detection. Through these mechanisms, ESCAViT effectively integrates spatial and spectral EEG features, outperforming traditional methods in capturing inter-lead dependencies.

### 3.4   Domain Robust Technique

ESCAViT integrates two domain-adaptive learning strategies, AES-Mix and LIGCL, to address data ambiguity, HOC issues, and class imbalance in EEG

**Table 1.** Comparison of EEG models on IIIC classification performance. 1D models are trained on raw EEG data, while 2D models use EEG spectrograms as input features.

| Input | Model Type | Method | ACC | F1 | KLD | TPS | mACC |
|-------|-----------|--------|-----|-----|-----|-----|------|
| **1D** | Graph | 1D GNN-CNN [19] | 0.394 | 0.246 | 0.830 | 0.192 | 0.248 |
|  | Transformer | EEG Conformer [20] | 0.351 | 0.298 | 0.869 | 0.337 | 0.341 |
|  | 1D-based | SPaRCNet [21] | 0.636 | 0.546 | 0.698 | 0.447 | 0.511 |
| **2D** | Transformer | AST(Tiny) [15] | 0.503 | 0.421 | 0.766 | 0.378 | 0.411 |
|  |  | w/ DRT | 0.554 | 0.462 | 0.741 | 0.453 | 0.457 |
|  | Domain-Adaptive Learning | DANN [22] | 0.495 | 0.280 | 0.816 | 0.163 | 0.284 |
|  |  | w/ DRT | 0.659 | 0.556 | 0.714 | 0.539 | 0.552 |
| **3D** | 3D CNN | ResNet3D [23] | 0.670 | 0.600 | 0.670 | 0.576 | 0.592 |
|  |  | w/ DRT | 0.744 | 0.684 | 0.623 | 0.692 | 0.685 |
|  | Ours | ESCAViT(base) | 0.719 | 0.662 | 0.628 | 0.663 | 0.653 |
|  |  | ESCAViT(w/ DRT) | **0.758** | **0.714** | **0.605** | **0.700** | **0.704** |

classification. Each technique targets specific challenges through complementary mechanisms.

AES-Mix addresses class imbalance by selectively augmenting minority classes (LPD, GRDA, LRDA) to resolve feature learning failures in underrepresented patterns [17]. Since RDA exhibit diagnostic features in 1-4Hz band, mixing is restricted to this range to preserve critical characteristics [18].

LIGCL targets data ambiguity from low inter-rater agreement through adaptive contrastive learning. It uses mixup ratio $\lambda$ as weights—higher for original-like samples to maintain boundaries, lower for mixed samples to control ambiguity.

Their synergistic integration overcomes individual limitations: AES-Mix alone dilutes majority class features while LIGCL alone over-sharpens minority class boundaries. Combined, they enable robust performance on imbalanced and ambiguous IIIC patterns.

## 4 Experimental Results

### 4.1 Dataset and Experimental Setup

We used the publicly available harmful brain activity in electroencephalography (EEG) dataset (https://www.kaggle.com/competitions/hms-harmful-brain-activity-classification, [24]). The dataset consists of 17,089 EEG segments from 2,711 patients, annotated by 20 neurophysiology experts into five IIIC-related patterns (Seizure, LPD, GPD, LRDA, GRDA) and an Other category.

The dataset exhibits severe class imbalance, with the Other class comprising 42% while LRDA accounts for only 5%. Expert agreement varies significantly across patterns, with LRDA and GPD showing the lowest consensus (0.73 and 0.80, respectively), indicating high inter-observer variability. These imbalances and ambiguities highlight the necessity for domain-adaptive learning strategies.

For the experimental setup, we divided the 17,089 samples into training, validation, and testing sets using stratified splitting in an 8:1:1 ratio. All models were trained on one NVIDIA GeForce RTX A6000 48GB GPU using the AdamW optimizer with a learning rate of 1e-4, weight decay of 1e-3, batch size of 8, and 50 epochs.

### 4.2   Baseline Models and Evaluation Metrics

ESCAViT was evaluated against five types of state-of-the-art EEG classification models. These include 1D-based approaches (SPaRCNet [21]), graph-based models (GNN-CNN [19]), Transformer-based architectures (EEG Conformer [20], AST [15]), domain-adaptive learning (DANN [22]), and 3D CNN (ResNet3D [23]).

$$\text{TPS} = \frac{\sum_i \mathbb{1}(\text{pred}_i = \text{true}_i) \cdot \mathbb{1}(\text{class}_i \neq \text{other})}{\sum_i \mathbb{1}(\text{class}_i \neq \text{other})} \tag{1}$$

To assess model performance, we used multiple metrics including Accuracy (ACC), macro-averaged F1-score, Mean Accuracy per Class (mACC), Target Pattern Sensitivity (TPS), and KL Divergence (KLD). The mACC [25] measures per-class accuracy to mitigate majority-class bias effects. TPS (Eq. 1) evaluates classification accuracy excluding the majority Other class, focusing on IIIC-related patterns. These two metrics served as primary indicators for evaluating classification performance under class imbalance.

### 4.3   Performance Evaluation and Comparative Analysis

As shown in Table 1, conventional 1D and 2D models struggle with IIIC pattern classification due to their limited ability to capture lead symmetry and seizure-specific patterns. Even SpaRCNet, a model specialized for IIIC, shows prediction bias with mACC and TPS below 51%. ResNet3D improves performance by leveraging spatial correlations but remains suboptimal due to its limitations in long-range feature extraction. In contrast, ESCAViT effectively captures inter-lead features through Pairwise Attention and ViViT-based architecture while extracting local features of seizure patterns via AST transfer learning, Lead Attention, and Convolutional Projection. This results in average performance improvements of 25.6% in F1, 30% in TPS, and 25.2% in mACC, demonstrating its ability to effectively capture seizure patterns while learning inter-lead correlations.

DRT further improves ESCAViT, achieving over 70% in mACC and TPS, confirming its effectiveness in mitigating prediction bias. Notably, DRT also enhances other spectrogram-based models, with DANN achieving a 16% accuracy increase. This demonstrates DRT's robustness in handling data ambiguity and class imbalance.

### 4.4   Visualization of Model Behavior

We analyzed model behavior using confusion matrices, T-SNE visualizations, and cluster-separation bar plots (Fig. 2). Compared to SpaRCNet, ESCAViT

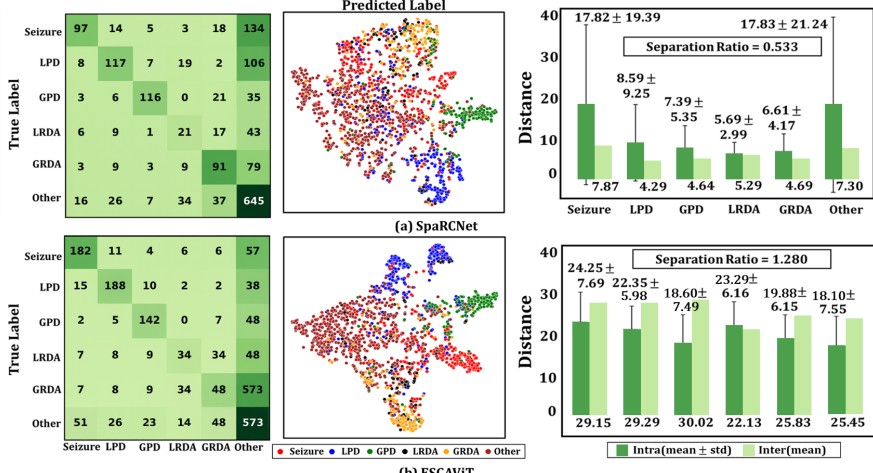

**Fig. 2.** Quantitative and qualitative comparison of models. (a) SpaRCNet from Table 1, (b) ESCAViT Base. Each model shows confusion matrix, t-SNE projection, and cluster-separation bar plot. Bar plots report mean intra-class distance (dark bars, $\pm$ 1 SD), mean inter-class distance (light bars), and separation ratio (inter/intra distance). Separation Ratio > 1 indicates well-separated clusters.

significantly reduced misclassifications into the Other class ($397 \rightarrow 187$) and achieved 31.4% improvement in seizure classification through Lead Attention and Convolutional Projection. Pairwise Attention enhanced inter-lead symmetry modeling, reducing LRDA-GRDA misclassification errors from 26 to 19.

The superior cluster separation performance of ESCAViT was confirmed through T-SNE visualizations and bar plots. SpaRCNet exhibited high variance in intra-class cohesion (intra metric), while ESCAViT achieved smaller Intra-Distance than Inter-Distance for all classes except LRDA, demonstrating high intra-class cohesion. ESCAViT achieved a separation ratio 0.747 points higher than SpaRCNet.

Occlusion sensitivity analysis [26] (Fig. 3) confirmed ESCAViT's ability to capture hemispheric relationships. Unlike ResNet3D, which fails to consider left-right correlations, ESCAViT effectively highlights symmetrical EEG features, improving classification of ambiguous patterns.

### 4.5 Ablation Study

To confirm that the interaction between AES-Mix and LIGCL is essential in the proposed DRT, we conducted experiments by individually applying each technique to ESCAViT (base). As shown in Table 2, individual application of each technique resulted in performance degradation. When only AES-Mix was

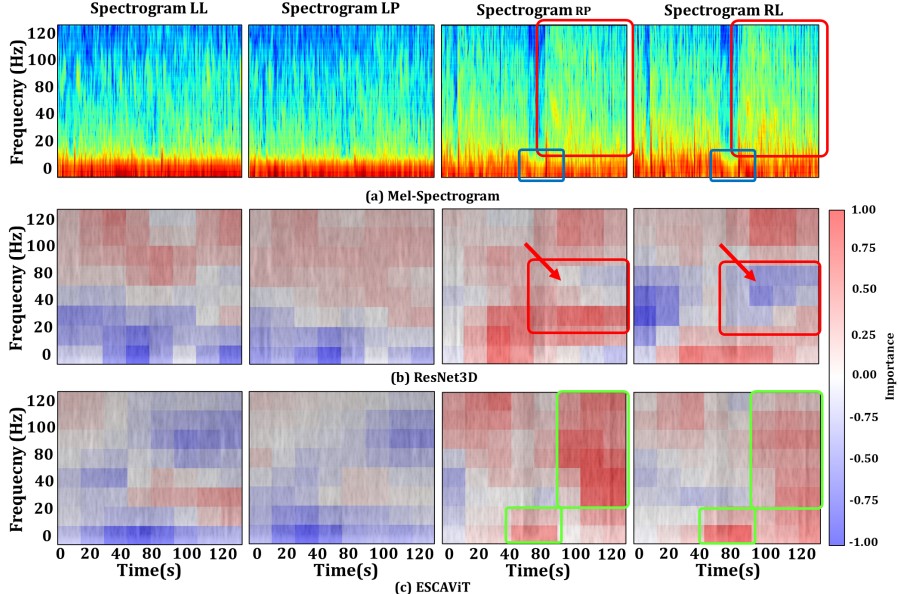

**Fig. 3.** Occlusion sensitivity analysis (patch size: 32×32, stride: 16×16). (a) Mel-Spectrogram visualization of an LRDA sample. Black boxes indicate regions where frequency and amplitude differences are observed between left and right channels. Right and left signals show similar patterns within each hemisphere. (b) ResNet3D and (c) ESCAViT feature importance heatmaps. Red boxes show ResNet3D assigning different importance values to the same time-frequency regions within right signals. Green boxes highlight ESCAViT assigning consistent importance, effectively capturing inter-lead symmetry.

applied, insufficient representation of the Other class led to approximately a 5.4% decrease in majority class accuracy. Conversely, applying only LIGCL caused excessive boundary sharpening, resulting in a 5.2% decline in mACC score. This demonstrates that only the combined application of both techniques in DRT can effectively address class imbalance and enhance model performance.

## 5   Conclusions

In this paper, we propose ESCAViT, a ViViT-based model for IIIC pattern classification. To enhance inter-lead correlation learning, we introduce AST Transfer Learning, Convolutional Projection, and Lead Attention mechanisms, along with a DRT combining AES-Mix and LIGCL to address data ambiguity and class imbalance. Experimental results demonstrate that the proposed model achieves superior performance in classification and seizure pattern recognition compared

**Table 2.** Performance comparison of individual and combined application of AES-Mix and LIGCL in the proposed DRT. DRT represents the combined use of both techniques, while w/o AES-Mix indicates LIGCL only and w/o LIGCL indicates AES-Mix only. OthACC represents the accuracy of the Other class, which is the majority class.

| Method | ACC | F1 | KLD | TPS | mACC | OthACC |
|---|---|---|---|---|---|---|
| DRT | **0.758** | **0.714** | **0.605** | **0.700** | **0.704** | **0.795** |
| w/o AES-Mix | 0.707 | 0.654 | 0.639 | 0.683 | 0.666 | 0.764 |
| w/o LIGCL | 0.723 | 0.663 | 0.628 | 0.692 | 0.652 | 0.741 |

to existing approaches. DRT effectively mitigates class imbalance and strengthens model generalization across diverse EEG patterns. ESCAViT shows how domain-adaptive modeling of biosignals can enhance the reliability of clinical decision support tools and contribute to developing generalizable AI systems suitable for real-world clinical deployment. For future work, we plan to validate the model's robustness on large-scale clinical datasets including long-term ICU recordings.

**Acknowledgments** This work was supported by the Institute of Information & Communications Technology Planning & Evaluation (IITP) grant funded by the Korea government (MSIT) [No. RS-2022-II220641, XVoice: Multi-Modal Voice Meta Learning], [No. RS-2022-00155915, Artificial Intelligence Convergence Innovation Human Resources Development (Inha University)].

**Disclosure of Interests** The authors declare that they have no conflicts of interest.

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
