# OpenReview forum: "ESCAViT: Symmetry-Aware EEG Classification"
_MICCAI.org/2025/Workshop/MSB_EMERGE — MSB EMERGE 2025 Oral_

### Official Review · Reviewer_5Xq9 · 2025-07-09

**Recommendation:** 4
**Confidence:** 3

**Clarity:**

The paper is clear and well-written, with minor areas for improvement in clarity

**Feedback:**

The paper introduces ESCAViT, a symmetry-aware, ViViT-based EEG classification model targeting IIIC pattern recognition. While the approach shows promising empirical results and introduces useful domain-specific adaptations, several aspects could be significantly improved to strengthen the paper's clarity:
(1) The motivation around class imbalance and expert disagreement is mentioned early, but lacks clarity - explicitly define the class distribution in the introduction.
(2) Several important implementation details are either omitted or unclear - specify architectural parameters - examples: ViViT layers, patch size, input resolution, positional encoding type, etc.
(3) Justify design choices, isolate and evaluate individual components - the contribution of most architectural modules beyond AES-Mix and LIGCL is not clearly demonstrated.
(4) Evaluate generalization and robustness - All experiments are conducted on a single dataset. The model's generalizability should be addressed on another public EEG dataset and report statistical significance or confidence intervals for key metrics.
(5) (Optional) Discuss model complexity — for example, how many parameters does each model have?

**Justification:**

ESCAViT tackles key challenges in IIIC EEG classification—specifically, inter-lead dependencies and class imbalance. The model shows consistent performance gains across well-chosen metrics, supported by evaluation tools such as confusion matrices and t-SNE plots. However, core components like LIGCL and AES-Mix are insufficiently described and lack formalization, while key architectural details are omitted. Additionally, the evaluation is limited to a single dataset, with no analysis of generalization or robustness. Overall, the paper offers a promising and practical solution, particularly if revisions enhance methodological clarity and provide stronger justification for the design choices.

**Reproducibility:**

Sufficient amount of details available for reproducing the main results, and open access is provided (or promised upon acceptance) to source code and/or data

**Strengths:**

Clear motivation: The paper tackles two key challenges in IIIC EEG classification: the difficulty in modelling hemispheric lead symmetry, crucial for seizure interpretation, and a highly imbalanced dataset where rare but important patterns like LRDA are overshadowed by dominant classes, complicating reliable classification.

Novel architectural approach: To address these challenges, ESCAViT integrates several key components:
(1) Lead and Pairwise Attention to capture inter-lead dependencies.
(2) A multi-stream setup combining a ViViT backbone with hemisphere-specific 3D ResNet branches for detailed feature learning.
(3) A domain-specific training strategy using AES-Mix and LIGCL to mitigate class imbalance and label ambiguity.

Performance and evaluation: ESCAViT outperforms 1D, 2D, and 3D baselines, with notable gains in F1, mACC, and TPS. The evaluation includes confusion matrices, t-SNE plots, occlusion maps, and an ablation study confirming the impact of AES-Mix and LIGCL.

**Summary:**

The paper proposes ESCAViT, a Transformer-based multi-stream framework designed to classify Ictal-Interictal-Injury Continuum (IIIC) EEG patterns in ICU patients. It addresses two main challenges in EEG classification: (1) Capturing inter-lead relationships and spatial dependencies, especially hemispheric symmetry. (2) Class imbalance and data ambiguity, which degrade classification accuracy. The key components of ESCAViT are: (1) Lead-Aware Feature Extraction: Includes Lead Attention and Pairwise Attention to model inter-lead dependencies. (2) Domain Robust Techniques for handling class imbalance and OOD generalization. (3) Multi-Stream Architecture enable fine-grained seizure pattern detection while preserving global EEG structure.

**Weaknesses:**

Unclear presentation of class imbalance: The abstract and introduction mention class imbalance, but details on its extent and impact are missing upfront. Key statistics only appear later in Section 4.1, weakening the early motivation.

No evaluation of generalization: The model is evaluated solely on a single dataset, with no external validation to assess generalizability. There’s also no cross-validation or statistical analysis to support claims of significant improvement.

Insufficient detail on method components: Key elements like the design of LIGCL and the role of Mixup ratios are described vaguely, with no clear explanation of how they function within the pipeline or what values and settings were used.

Unclear architectural specifications and contributions: The paper provides little justification for architectural choices such as the use of ViViT, its depth, patch size, or input dimensions. Beyond the ablation of AES-Mix and LIGCL, the impact of other components is not clearly isolated or systematically assessed.

---

### Official Review · Reviewer_aivr · 2025-07-09

**Recommendation:** 3
**Confidence:** 4

**Clarity:**

The paper is clear and well-written, with minor areas for improvement in clarity

**Feedback:**

1. Please number the references in the order they appear in the text.
2. Once “out-of-distribution (OOD)” is introduced, use only the abbreviation “OOD” for subsequent mentions.
3. After introducing “Domain Robust Technique”, be sure to add the abbreviation “DRT” directly after the full term on first use.
4. The description “mixing operations are selectively applied to specific frequency bands” is central to understanding AES-Mix, yet the paper does not explain what criteria are used for this selection. This should be clarified.
5. The phrase “Values > 1 indicate well-separated clusters” should be revised to “Separation Ratio > 1 indicates well-separated clusters” for clarity.
6. In Fig. 3, the two green boxes should be applied to the same region to ensure valid visual comparison.
7. As mentioned in the weaknesses, the paper should at least define what kind of OOD it addresses. Furthermore, it should include specific experimental results related to this claimed OOD handling, rather than treating it vaguely.
8. The paper could be strengthened by including some mathematical formulations or theoretical reasoning rather than relying solely on hypotheses and empirical results. For example, demonstrating how AES-Mix and LIGCL mathematically complement each other would provide helpful theoretical backup.

**Justification:**

The objective of this study is meaningful, and the paper is well-structured. However, across the entire methodology, there is a lack of explanation as to how each module actually functions. Several modifications should be made for the paper to be accepted.

**Reproducibility:**

Sufficient amount of details available for reproducing the main results, and open access is provided (or promised upon acceptance) to source code and/or data

**Strengths:**

1. Figures are easy to comprehend.
2. Idea of splitting left/right hemispheres into an independent pathway is persuasive in that it is consistent with clinical interpretation methods.
3. Objective of the paper is well defined and the key contributions are clearly structured.

**Summary:**

The paper proposes ESCAViT, a novel Transformer-based framework for classifying EEG patterns in the Ictal-Interictal-Injury Continuum (IIIC), which is critical for neurological assessment in intensive care units. ESCAViT addresses key challenges in EEG analysis, including inter-lead symmetry and class imbalance. The model introduces symmetry-aware mechanisms such as Lead Attention and Pairwise Attention to capture spatial relationships between EEG leads, along with a multi-stream architecture utilizing hemisphere-specific 3D ResNet branches. To enhance generalization and robustness, the authors propose a Domain-Robust Technique (DRT), integrating AES-Mix, a data augmentation method, and LIGCL, a contrastive learning strategy guided by lead relationships. Experiments show that ESCAViT outperforms existing methods.

**Weaknesses:**

1. The Method section emphasizes the strengths of ESCAViT but lacks rigorous explanation of how the proposed components concretely address the stated problems. Rather than clearly linking AES-Mix and LIGCL to their corresponding challenges (e.g., class imbalance, data ambiguity), the paper presents an all-encompassing claim that ESCAViT addresses class imbalance, OOD, and data ambiguity. These should be explained in a one-to-one manner.
2.  The paper mentions inter-observer variability as a key source of data ambiguity but fails to address it directly. Instead of modeling label uncertainty or annotator disagreement explicitly (e.g., through soft labels, confidence-weighted loss, or uncertainty modeling), the authors implicitly suggest that a more robust deep learning model suffices.
3. The claim that ESCAViT handles OOD scenarios is unsubstantiated. There is no definition of what constitutes “out-of-distribution” in the context of EEG in this study. In medical imaging, OOD typically refers to differences in data distribution arising from scanner variation or shifts in patient demographics—none of which are discussed or tested in this work.
4. While the authors claim to address the class imbalance problem through the proposed DRT module, they use Target Pattern Sensitivity (TPS) as a core evaluation metric. However, TPS explicitly excludes the majority “Other” class from its calculation, which undermines its relevance for assessing class imbalance mitigation.

---

### Official Review · Reviewer_3s8L · 2025-07-10

**Recommendation:** 4
**Confidence:** 2

**Clarity:**

The paper is generally clear but has some clarity issues that could be addressed with moderate revision

**Feedback:**

- As mentioned above, Fig. 1 is a major issue for me. It should be improved for publication.
- I am not sure if this is a common concept in EEG, but I am missing some introduction into why lead symmetry is so important. Also, the occlusion sensitivity experiment falls a bit short.
- Shorten the running title
- I think you can make Fig. 2 a bit wider, which will render the Figure less crowded

**Justification:**

I am not so confident in EEG classification, but I generally feel like this paper proposes a great architecture and training scheme for EEG classification. The comparisons with baseline models seem sound, clearly showing that ESCAViT beats those.

**Reproducibility:**

Sufficient amount of details available for reproducing the main results, and open access is provided (or promised upon acceptance) to source code and/or data

**Strengths:**

- ESCAViT clearly outperforms a large number of baselines methods

**Summary:**

The authors propose a novel architecture and training scheme for EEG classification

**Weaknesses:**

- Fig. 1 is difficult to understand. Due to multiple reasons:
	1. Several symbols are used but not introduced, e.g. G, S
	2. Fire emoji usually means that certain layers are trainable. Do you have frozen parameters? What does the emoji mean in your Figure?
	3. ImageNet, Audio Spectogram, Kinetics400v1 indicates that the layers were pre-trained? If so, I would probably leave it away as this is too detailed.
	4. In the left (a, b), you use boxes to highlight specific parts, but in the right (c, d), you use curly braces. I suggest standardizing it.
- I do not understand the lead and pairwise attention parts. Are the inputs condensed to just 4 values?
- What is the time frame of each sample?